# Gut Barrier Damage and Gut Translocation of Pathogen Molecules in Lupus, an Impact of Innate Immunity (Macrophages and Neutrophils) in Autoimmune Disease

**DOI:** 10.3390/ijms23158223

**Published:** 2022-07-26

**Authors:** Awirut Charoensappakit, Kritsanawan Sae-khow, Asada Leelahavanichkul

**Affiliations:** 1Center of Excellence in Translational Research in Inflammation and Immunology (CETRII), Department of Microbiology, Faculty of Medicine, Chulalongkorn University, Bangkok 10330, Thailand; awirut.turk@gmail.com (A.C.); kritsanawan_29@hotmail.com (K.S.-k.); 2Nephrology Unit, Department of Medicine, Faculty of Medicine, Chulalongkorn University, Bangkok 10330, Thailand

**Keywords:** leaky gut, innate immunity, systemic lupus erythematosus

## Abstract

The gut barrier is a single cell layer that separates gut micro-organisms from the host, and gut permeability defects result in the translocation of microbial molecules from the gut into the blood. Despite the silent clinical manifestation, gut translocation of microbial molecules can induce systemic inflammation that might be an endogenous exacerbating factor of systemic lupus erythematosus. In contrast, circulatory immune-complex deposition and the effect of medications on the gut, an organ with an extremely large surface area, of patients with active lupus might cause gut translocation of microbial molecules, which worsens lupus severity. Likewise, the imbalance of gut microbiota may initiate lupus and/or interfere with gut integrity which results in microbial translocation and lupus exacerbation. Moreover, immune hyper-responsiveness of innate immune cells (macrophages and neutrophils) is demonstrated in a lupus model from the loss of inhibitory Fc gamma receptor IIb (FcgRIIb), which induces prominent responses through the cross-link between activating-FcgRs and innate immune receptors. The immune hyper-responsiveness can cause cell death, especially apoptosis and neutrophil extracellular traps (NETosis), which possibly exacerbates lupus, partly through the enhanced exposure of the self-antigens. Leaky gut monitoring and treatments (such as probiotics) might be beneficial in lupus. Here, we discuss the current information on leaky gut in lupus.

## 1. Introduction

Systemic lupus erythematosus (SLE or lupus) is a common autoimmune disease with the involvement of multiple organs (skin, joints, and central nervous system) due to the deposition of the immune complexes between self-antigens and autoantibodies in the microcirculatory networks of several organs [1]. There is multifactorial pathogenesis of lupus consisting of the combination between genetic predispositions and environmental factors that trigger abnormal immune responses in the innate (antigen-presenting cells) and adaptive immunity (self-reactive T and B cells) [1]. While the hallmark of SLE is an abnormality in the adaptive immunity resulting in the elevation of autoantibodies (mostly against the nuclear antigens), the influence of innate immunity-induced inflammation in lupus disease progression is well known [1]. Although the gastrointestinal (GI) symptoms in lupus are not predominant, the immune complex deposition in the gut of patients and mice with lupus is mentioned, partly due to the large surface area of the gastrointestinal system [2]. Interestingly, the gut immune complex deposition induces inflammatory responses and possibly facilitates gastrointestinal permeability defect with the translocation of pathogen molecules from the gut into the blood circulation, referred to as “gut leakage or leaky gut” [2], because (i) the reactions against pathogen-associated molecular patterns (PAMPs) that are foreign to the host are usually more severe than the responses toward the host antigens [3], (ii) the presence of PAMPs in blood possibly induces a potent immune activation, especially innate immune responses [2], and (iii) acute and chronic inflammation is an exacerbating factor of lupus activity [4]. The translocation of PAMPs or viable organisms from the gut might be an endogenous factor to trigger inflammation or disease flare-ups in patients with lupus. Interestingly, a defect of gut permeability could be demonstrated without any symptoms or only subtle non-specific symptoms (fatigue, nausea, and bloating) [5]. Hence, the flare-up of lupus activity due to the inflammatory responses against a silent gut translocation of endogenous PAMPs might be responsible for the fluctuation of lupus disease activity in some patients without an obvious exposure to the exogenous exacerbating factors. However, the concerns about these silent exacerbation factors in lupus are still too few despite the established methods for the determination of gut leakage. It is interesting to note that patients with lupus are not only susceptible to the exogenous environmental factors (chemical substances, organisms, and PAMPs) [6], but also the stimulators from the endogenous factors due to gut translocation. In this review, we discuss the possible impacts of leaky gut in SLE focusing on the activation of innate immunity and proposed the possible treatment of gut leakage based on the improved understanding of this topic.

## 2. General Aspects of Gut Permeability Defects

In each person, the gastrointestinal epithelial lining approximately includes the surface area of a small room (32 m^2^), and not only separates the host from the external environments but functions as a first-line innate immune defense to safeguard the entry of foreign antigens [2]. Interestingly, this barrier consists of only a single layer of epithelial cells with cell junctional complexes and several humoral factors, including mucins, antimicrobial molecules, immunoglobulins, and cytokines [7,8]. Under normal conditions, the intestinal barrier acts as a barrier to select the entry of only some molecules mainly with two pathways: the intestinal cell paracellular space and a transcellular passage [2]. The translocation through the paracellular space is regulated by cell junctional complexes, including tight junctions (TJs), adherens junctions, and desmosomes [2]. The epithelial TJs, located at the apical end of lateral membranes, consist of four major protein complexes; occludin, claudin, junction-adhesion molecules (JAMs), and the coxsackievirus-adenovirus receptor (CAR) protein, which regulate the paracellular trafficking and allow the molecules that are smaller than 3.6 A° (or 0.6 kDa) to pass through the normal paracellular passage [2]. This size selectivity of the gut barrier prevents gut translocation of viable organisms and several PAMPs which are the drivers of systemic inflammation [8,9,10,11]. As such, Gram-negative bacteria and fungi (mostly *Candida albicans*) are the most and the second most prominent organisms in the human gut and are the main source of lipopolysaccharides (LPS) and (1→3)-β-d-glucan (BG), respectively, in the intestinal contents [2]. Both LPS, molecular weight (MW) ranging from 10 to >100 kDa, and BG (MW 6 to >600 kDa) are the major cell wall components of bacteria and fungi which might be the main PAMPs in gut contents with potent immune activation properties [8,9,10,11]. Moreover, the free nucleic acid from the microbial breakdown might be another pathogen molecule from the gut translocation that may activate immune responses. The microbial-free DNA is rapidly naturally broken down by several processes (depurination or deamination DNA) into smaller sizes of less than 100 bp (65 kDa) which can cross the gut barrier [12,13] even though the intact bacterial genome at a molecular size of 100 to 15,000 kbp (6.5 × 10^4^–9.8 × 10^6^ kDa) [14] cannot pass through the gut barrier. The detection of microbial-free deoxyribonucleic acid (DNA) in several conditions without positive bacteremia; such as coronavirus disease (COVID-19) and obesity, and the detection of intestinal bacterial ribosomal RNA (rRNA) in the lungs of patients with acute respiratory distress syndrome (ARDS), and leaky gut measurement in patients with the post-abdominal operation, imply gut translocation of microbial nucleic acids [9,15,16,17]. During gastrointestinal translocation, PAMPs could enter the body’s circulatory system through the lymphatics (mesenteric lymph nodes and thoracic lymph duct) and portal circulation (portal vein) [2]. Furthermore, the attachment of LPS to lipoproteins/chylomicrons from the gut during routine lipid absorption may physiologically enhance LPS levels in systemic circulation [18]. Because the gut barrier consists of only a single layer of enterocytes in a very large surface area, silent gut leakage without the detectable PAMPs in the blood circulation might be a normal physiologic condition [19]. Indeed, several host mechanisms regulate PAMPs from gut translocation, mostly from neutralization using innate immunity, for example, the LPS binding protein, soluble cluster of differentiation (CD)-14, cell surface receptors (toll-like receptor (TLR)-4, dectin, and scavenger receptors), complement receptor 3, and some enzymes such as acyl-oxy-acyl hydrolase (AOAH), alkaline phosphatase, and DNAase or RNAase [2,20,21]. Hence, the detection of PAMPs in blood indirectly indicates gut translocation of a significant abundance of PAMPs beyond the capacity of host neutralization mechanisms possibly due to the significant intestinal mucosal damage despite asymptomatic clinical presentation in some cases.

## 3. Gut Permeability Defects in Lupus

In lupus, the immune complex in the vascular networks is the main multi-organ pathogenesis that can deposit throughout the gut from mouth to rectum, due to the enormous surface area of the organ, in different intensities, including asymptomatic deposition, mucosal injury, or bowel ischemic vasculitis [22,23]. The autopsy and endoscopic studies exhibited abdominal injury in approximately 70% of patients with SLE, although only 10% of these patients show GI symptoms [24,25]. Indeed, the immune-complex deposition in the GI tract during the active lupus nephritis without the observed clinical GI abnormalities (diarrhea or weight loss) in the lupus rodent model from the deletion of Fc gamma receptor IIb (FcgRIIb-/- mice) [26,27] with the defects on intestinal tight junctions (Figure 1) are demonstrated. Notably, FcgRIIb is the only inhibitory receptor among the Fc gamma receptor (FcgR) family, and the loss of FcgRIIb facilitates immune responses resulting in the spontaneous development of full-blown lupus nephritis (serum anti-dsDNA, proteinuria, and uremia) after 6 months in FcgRIIb-/- mice.

Despite the non-GI symptoms, gut leakage during active lupus is indicated by the elevation of PAMPs in serum (LPS and BG) and the tight junction protein level in blood circulation in either patients or mice with lupus [26,28,29,30,31]. Moreover, gut permeability defects might be severe enough for the translocation of viable bacteria. As such, gut translocation of *Enterococcus gallinarum*, one of the pathobiont in some situations, to the liver and other organs trigger anti-dsDNA (the major autoantibody in lupus) through TLR-7/8 activation in New Zealand White (NZW)/F1 lupus mice (and also possibly in patients), which is attenuated by antibiotics, has been demonstrated [28]. Nevertheless, gut permeability damage in lupus is not only due to immune-complex deposition in the gut, but also from the adverse effects of some medications used in autoimmune diseases, including nonsteroidal anti-inflammatory drugs (NSAIDs), corticosteroids, and disease-modifying antirheumatic drugs (DMARDs) [32,33,34,35]. As such, the long-term use of NSAIDs induces GI lesions (in the stomach and intestines) [33,36,37] and disrupts gut integrity partly through cell apoptosis induction [34], which possibly exacerbates lupus progression in a lupus mouse model [27]. It is interesting to note that gut translocation of PAMPs after NSAIDs use possibly facilitates lupus activity through a significant systemic inflammation as indicated by increased serum cytokines, despite the anti-inflammatory property of NSAIDs [27]. Similarly, the induction of gut leakage by a low dose of dextran sulfate solution (direct damage to enterocyte tight junction) induces lupus disease activity without diarrhea [26], implying that the silent leaky gut without GI symptoms is a possible exacerbating factor of lupus. Additionally, infection, including intestinal and systemic infection, is a major complication in patients with lupus due to immunosuppressive drugs [38,39,40,41] or defects in the pathogen control from lupus itself, as the infection is a common cause of death even before the era of immunosuppressive drugs [42,43,44]. Perhaps a possible leaky gut during enterocolitis or diarrhea in patients with lupus should be a concern. While gut translocation of PAMPs in intestinal infection is easily explained through the enterocyte tight junction injury, leaky gut in systemic infection is possibly a result of systemic inflammation-induced enterocyte damage [2]. Indeed, diarrhea in some patients with systemic infection (primarily non-gastrointestinal infections) [45] and endotoxemia in severe viral infection [17,46] support a possibility of leaky gut during systemic infection. Although diarrhea is not a good biomarker for leaky gut, as diarrhea is not demonstrated in sepsis or asymptomatic *Clostridium difficile* infection in mouse models [47], diarrhea in lupus might be associated with leaky gut and, at least, a subtle systemic inflammation. Despite the well-known systemic inflammation during active lupus (especially endotoxemia without systemic infection), data on the consequence of gut leakage as an endogenous exacerbation factor of lupus is still lacking. Hence, all of the evidence mentioned above suggests that leaky gut in lupus is possible and might be clinically important.

## 4. Gut Dysbiosis in Lupus

Several studies demonstrated that the healthiness of intestinal mucosa [48,49] and the proper immune maturation [50,51,52,53] depended on gut microbiota [49,54,55]. Germ-free mice show decreased secreting IgA plasma cells with the thinner mucus layer and Peyer’s patches when compared with regular mice [56,57]. An increase in pathogenic organisms (pathobionts) in the gut, referred to as “gut dysbiosis”, is one of the possible causes of leaky gut [58] that are demonstrated in several diseases, including obesity, inflammatory bowel disease (IBD), and autoimmune diseases [59,60,61]. Notably, gut dysbiosis can be a cause or a consequence of gut leakage. As such, oral administration of bacteria or fungi caused leaky gut from an increase in pathobionts [62,63], while leaky gut by dextran sulfate induces dysbiosis [26]. Gut dysbiosis in SLE might be a result of a combination of (i) the genetic defects in the immune responses against some gut organisms that selectively allow some group of organisms to survive in the gut, together with (ii) the gut environmental factors from the host behaviors (diets, alcohol, smoking, and medications). For example, fecal transfer from triple congenic lupus-prone mice during active lupus to germ-free mice induces autoantibodies and autoimmune phenotypes [64]. Lupus caused by complement deficiencies (C1q, C2, C3, and C4) [65,66] might be associated with gut dysbiosis due to a defect in the control of some organisms as indicated by the dysbiosis in C3 deficient mice [67]. However, the environmental factors are more important than host genetics in the alteration of gut microbiota from a study of 1,046 healthy individuals [68].

Nevertheless, the increased *Bacteroidetes* with decreased *Firmicutes* was demonstrated in patients with lupus in different regions of the world [69,70,71,72], and in various lupus models [28,31,58,71,73,74]. Indeed, *Firmicutes*, mostly Gram-positive bacteria with obligate aerobes or facultative anaerobes, are the major gut microbes that alter complex carbohydrates into short-chain fatty acids (SCFA), particularly butyrate, which serve as growth factors for gut epithelium [75]. Meanwhile, *Bacteroides*, mostly Gram-negative anaerobes, are pathogenic bacteria in some situations [76]. Normal gut microbiota also maintains gut epithelial integrity using other substances. For example, tryptophan metabolites (indole and tryptamine) upregulate *Muc2* (mucin production), through the pregnane X receptor (PXR) and aryl hydrocarbon receptor (AhR), and induce glucagon-like peptide 1 (GLP-1; an enterocyte growth factor) from enteroendocrine L-cells [77,78]. Leaky gut with reduced *Muc2* in PXR-deficient mice has been demonstrated [79,80]. Although the *Firmicutes/Bacteroides* ratio is increased and decreased in IBD and obesity, respectively, the clinical use of this ratio, including in lupus, is still unclear. It is possible that inflammatory responses from leaky gut, possibly due to the reduced *Firmicutes*, might initiate or exacerbate lupus in some patients who are already prone to lupus, or gut inflammation during active lupus might decrease *Firmicutes* bacteria. Despite the unclear role of gut dysbiosis in lupus, healthy gut integrity prevents gut translocation of PAMPs that might possibly be helpful for the control of lupus exacerbation. Despite inconclusive pathogenesis, gut dysbiosis in lupus is well known and might be associated with leaky gut, which can be a cause or a consequence of lupus.

## 5. Leaky Gut-Induced Innate Immunity and Cell Death in Lupus

Innate immunity is the first-line recognition of PAMPs and damage-associated molecular patterns (DAMPs) from microbes and host cells, respectively. Although the abnormalities in the adaptive immunity are dominant in SLE as indicated by the immune-complex formation, lupus exacerbation by innate immunity-induced inflammation and innate immunity dysfunction in lupus is well known [1,81]. For example, the defect of macrophages in clearing apoptotic cell debris induces prolonged exposure to autoantigens [82,83] and abnormal adaptive immunity due to the antigen processing property of dendritic cells (DCs) in lupus [84,85,86]. Accordingly, LPS and BG from gut translocation mainly activate several innate immune cells through TLR-4 and dectin-1, respectively, and the co-presence of LPS and BG induces the synergy of pro-inflammatory responses. Additionally, there is a dramatic decrease in glomerular IgG deposition and mesangial cell proliferation with reduced autoantibody titers in TLR-2- or TLR-4-deficient MRL lymphoproliferation strain (MRL/lpr) lupus mice [87]. Despite a limited exploration of leaky gut in lupus models, gut dysbiosis and the correlation between gut microbiota and systemic inflammation in lupus is well established [58,88,89]. During the gut permeability defect in lupus, PAMPs and DAMPs enhance systemic inflammation and induce lupus disease progression via the activation of several innate immune cells including neutrophils, macrophages, and DCs. Despite the diverse pathogenesis of lupus, defect in inhibitory FcgRIIb is one of the causes of lupus and hypofunctional FcgRIIb polymorphisms are frequently reported in the population in Asia and East Asia partly due to the selection by malarial infection in the region [90]. Defect in inhibitory FcgRIIb causes hyper-responsiveness against malaria in these regions, while enhancing the incidence of lupus [90]. Because FcgRIIb is detectable in neutrophils, macrophages, DCs, and B cells (but not T cells), FcgRIIb-/- mice have been used to explore both innate and adaptive immune cells in lupus [91,92]. Because of the possible cross-talk between the Fc gamma receptor and TLR-4, which is a pattern recognition receptor against several ligands [46], both the endogenous human molecules (DAMPs; such as heat-shock protein, beta-amyloid, and cell-free DNA) and exogenous microbial or environmental molecules (such as LPS, mannan, and particle matter 2.5) [93], the co-presence of these TLR-4 ligands together with activation of the activating-FcgRs (non-FcgRIIb), perhaps by the natural immunoglobulin, might induce synergistic pro-inflammatory responses through the FcgR-TLR-4 cross-talk [94]. Although the FcgRIIb defect is only a part of the genetic abnormalities in lupus, FcgR-TLR-4 cross-talk might be responsible for the hyper-immune responses against several conditions in patients with lupus through macrophages and neutrophils, and the blockage of spleen-tyrosine-kinase (Syk, a shared downstream signaling molecule of both FcgR and TLR-4) attenuates lupus-induced inflammation both in mice and in patients [31,95]. This information supports the impact of innate immunity in lupus.

Furthermore, several processes of cell death, such as cell apoptosis and necrosis, are a common consequence of inflammation that could exacerbate lupus activity partly through the presentation of DAMPs, possibly due to inadequate clearance of these molecules during overwhelming cell death [96]. The hyper-inflammatory responses of immune cells (both innate and adaptive immunity) and the immune-induced cell injury in several organs by several stimuli (such as cytokines, PAMPs, and mitochondrial DNA) induce both apoptosis and necrosis of immune cells and parenchymal cells in lupus. Subsequently, the extracellular nucleic acids from host cells and mitochondria and nucleoproteins from the damaged cells might be the autoantigens that facilitate the autoantibody production and circulating immune-complex (CIC) deposition, which highlights the importance of cell death and DAMPs in lupus [97,98]. Although CIC deposition is a possible continuous process in lupus, an alteration in the abundance of CIC is possible which might be responsible for the variation of lupus activity (exacerbation versus inactivity) [99]. In lupus, the production of CIC and lupus exacerbation possibly depends on the increase in abundance of self-antigens from cell death [100,101] and autoantibody production, possibly through the non-specific activation of B cells (or plasma cells) by a viral infection, vaccination, or some medications [102,103]. For the enhanced self-antigen exposure, immune responses against pathogen molecules in blood during leaky gut might increase cell apoptosis as the spleen is responsible for the recognition of foreign molecules in the blood, and profound or chronic activation of some cells in the spleen (such as in the splenic marginal zone) may cause cell apoptosis [104]. For example, LPS during lupus-induced leaky gut might enhance spleen apoptosis, as demonstrated in LPS injection [105] and colitis mouse models [26]. The lupus exacerbation from the enhanced exposure to the self-antigens due to the increased apoptosis is also indicated in cancer treatment by anti-program cell death (PD)-1 (an apoptosis inhibitor) [106]. Moreover, a leaky gut also possibly facilitates autoantibody production. Accordingly, while infection and vaccination induce some specific clones of B cells using some growth factors, such as interleukin (IL)-2, and the pro-inflammatory processes (similar to the role of adjuvants), these factors might non-specifically stimulate self-reactive B cells clones in the host with lupus. Hence, the non-specific inflammatory responses against pathogen molecules during leaky gut might increase self-antigen exposure and accidentally facilitate autoantibodies.

Then, gut leakage that is severe enough to induce systemic inflammation might accelerate immune cell apoptosis, especially in spleens and livers, which exacerbates lupus activity [26] as demonstrated in the working hypothesis diagram (Figure 2). Likewise, cell death by neutrophil extracellular traps (NETs), referred to as “NETosis”, is a form of neutrophil death triggered by PAMPs (including LPS and BG) to release extracellular nuclear chromatin [107,108], which is a direct anti-organismal mechanism together with the stimulation of innate and adaptive immunities [101,109]. However, NETosis also induces the exposure of nuclear antigens that accelerate autoantibodies and lupus activity through several mechanisms, including (i) interferon (IFN) type I production from peripheral blood mononuclear cells and DCs through TLR-9 activation [109,110], (ii) the post-translational modifications (histone acetylation and citrullination) of some proteins, (iii) the impaired NET degradation through DNase deficiency [109,110], and (iv) the complex of anti-NET antibodies with NETs that block DNase and enhance the cells positive for TLR-9 and CD32 (DCs, monocytes, B cells, and GM-CSF-induced neutrophils) [96]. Because both LPS and BG from lupus-induced gut translocation induce NETosis that is possibly synergized by activating-FcgRs, gut leakage can enhance NETosis in lupus [111]. Moreover, pyroptosis [112], canonically dependent cell death through the plasma membrane pores using inflammasome-caspase 1-gasdermin-D that causes osmotic lysis, might also enhance the presentation of self-nucleic acids in lupus [113,114]. Pyroptosis also increases high-mobility group box 1 (HMGB1), a nuclear DNA binding protein ubiquitously expressed in eukaryotic cells, that increases in patients with SLE [115,116], and then, HMGB1 binding DNA complex can stimulate type I IFN release from DCs through TLR-9 [115,117] and activates B cells via the advanced glycation end-products (RAGE) receptor [118]. Hence, gut leakage in lupus possibly activates inflammatory responses in several types of immune cells severe enough to induce death of these immune cells and cells in several organs that are responsible for the exposure of self-antigens, autoantibody production, and lupus exacerbation.

## 6. Leaky Gut-Induced Inflammation and Molecular Mimicry in Lupus

Inflammation not only exacerbates lupus through the induction of cell death and the non-specifically enhanced autoantibody production but also mediates lupus activity through other pathways. Interestingly, some lupus-associated pathogenesis cytokines can be induced by pathogens or gut microbial translocation. For example, interferon (IFN) type I is an important cytokine that augments adaptive immunity through enhanced antigen presentation and promotes the high-affinity antigen-specific T and B cells in lupus pathogenesis [119]. It is also induced by CD11c positive DCs in the intestinal laminar propria during bacterial colitis [119,120,121,122] to prevent enterocyte disruption [123,124]. Additionally, extracellular DNAs exacerbate lupus activity through type I IFN activation from DCs in the intestinal laminar propria through TLR-9 [111]. Then, IFN type I during some bacterial colitis might be responsible for the lupus exacerbation [125]. Moreover, TNF-α and IL-6 are also important for lupus exacerbation as the potent pro-inflammatory cytokines in macrophage and neutrophil activation. As such, gene expression profiles in myeloid-derived macrophages from patients with SLE demonstrate increased gene expression of pro-inflammatory macrophages (M1 polarization), including signal transducer and activator of transcription (STAT)-1, suppressor of cytokine signaling (SOCS)-3, CD80, and CD86, and a decrease in the genes of alternatively anti-inflammatory macrophages (M2 polarization) (STAT-3, STAT-6, and CD163) [126,127,128]. Not surprisingly, studies in patients and lupus mice also exhibit an increasing baseline of TNF-α and IL-6 compared to healthy volunteers, which correlates with lupus disease severity [31,129,130]. In FcgRIIb-/- lupus mice, LPS from gut leakage in synergy with BG induces prominent M1 pro-inflammatory macrophages partly through the induction of TNF-α and IL-6 [131,132,133,134,135]. At the same time, TNF-α also directly induces programmed cell death including apoptosis and necroptosis, which accelerates the excess of auto-nuclear antigens [136]. Mice with a chronic overproduction of TNF-α (*TNF^deltaARE^* mice) spontaneously develop Crohn’s disease-like inflammation in the small intestine [137], suggesting that TNF-α may directly induce leaky gut. Due to the possible dual impacts in lupus pathogenesis and the responses against gut organisms and leaky gut of some cytokines, these cytokines might be responsible for the correlation between lupus exacerbation and intestinal inflammation. Furthermore, molecular mimicry is a process that non-self-peptides, mainly from microbial origins, share sequence homology with self-peptides, eliciting cross-reactive recognition between foreign and self-antigens, leading to activation of autoreactive T cells in several autoimmune diseases, including lupus [138]. Both DCs and macrophages are well known as professional antigen-presenting cells (APC) which are found in many tissues, along with the gastrointestinal tract. Interactions between APC and T cells are a direct link between innate and adaptive immunity through the process of antigens on major histocompatibility complexes (MHC) class II. Thus, organismal molecules from gut translocation might enhance the autoantibody production in lupus due to the molecular mimicry between some microbial and self-antigens. As such, the intestinal expansion of *Ruminococcus gnavus* (RG2strain) is associated with lupus nephritis along with the detection of antibodies against lipoglycans (a molecule in the cell wall of *R. gnavus*) with increased anti-dsDNA (a specific lupus antibody) and the extract of *R. gnavus* RG2 cross-reacts with anti-dsDNA antibodies, suggesting the molecular mimicry between RG2 cell wall moieties and native DNA molecules [73]. Additionally, the antibody against Ro protein (Ro60, a 60 kDa self-antigens) that is commonly found in lupus, cross-reacts with the Ebstein–Barr virus nuclear antigen-1 (EBNA-1), and the ortholog Ro60-containing is demonstrated in *Bacteroides thetaiotaomicron* (gut commensal bacteria) in microbiome analysis from patients with SLE [138,139]. These data suggest that lupus humoral autoimmunity might be initiated via molecular mimicry between PAMPs from gut translocation and self-antigens highlighting another correlation between gut permeability defect and lupus exacerbation or initiation.

## 7. Monitoring of Gut Leakage in Lupus

Interestingly, several methods might be suitable for the detection and monitoring of leaky gut in lupus. The direct measurement of gut integrity is based on (i) the detection of non-absorbable carbohydrates or other probed molecules in urine (or in the blood) after an oral administration [19] and/or (ii) the recognition of PAMPs in blood without an obvious source of infection, such as endotoxemia and glucanemia, as an indirect biomarker of gut translocation of pathogen molecules with large MW [47]. Although there are several molecular probes for leaky gut, such as (i) LPS detection assays, including Limulus amebocyte lysate (LAL), endotoxin activity assay (EAA), and ELISAs, (ii) BG tests, and (iii) an oral administration of a non-absorbable carbohydrate (before detection of the carbohydrate in blood or in urine) with several benefits and limitations [140,141], the use of these strategies for monitoring leaky gut in lupus is less common. Notably, the detection of serum zonulin (a regulator of epithelial and endothelial barrier functions) [142] is another possible indicator of gut integrity damage; however, the association between gut barrier defect and serum zonulin is still inconclusive [143,144,145]. Due to (i) the clinical availability and the convenience of the indirect leaky gut parameters (serum LPS and/or BG), and (ii) the safety of a single oral carbohydrate administration (mannitol with lactulose or sucralose), we propose to use these parameters for leaky gut monitoring in patients with lupus.

## 8. Therapeutics Targeting Gut Leakage in Lupus

Because gut dysbiosis, gut leakage, and the translocation of PAMPs might be associated with lupus disease activity, gut dysbiosis attenuation, gut integrity improvement, and the neutralization of PAMPs are potential methods for slowing SLE progression in susceptible individuals. Among several proposed interventions, probiotic administration and fecal transplantation are frequently mentioned [146]. Accordingly, probiotics are beneficial microbes that colonize the GI tract and confer beneficial effects on the host through several mechanisms including competition, anti-bacterial effects, protection of the GI barrier, and modulation of immune responses. The clinical implication of probiotics with the diverse bacterial species was firstly described in 1954 [147] in several diseases [146]. In SLE, several beneficial probiotics, particularly *Lactobacillus* spp., have been reported (Table 1). However, probiotics with mixed organisms are usually used, despite an uncertain benefit, over the single strain probiotic products [148]. Hence, fecal transplantation, an administration of feces from healthy donors that consist of several organisms, might be even more beneficial than multi-bacterial probiotics. Although fecal transplantation is never been tested in patients with lupus, the fecal microbiota transplantation (FMT) of feces from patients with lupus into germ-free mice induces some lupus-like characteristics, including increased autoantibodies, cytokines, altered immune cell distribution (in mucosa and blood), and upregulated SLE-related genes [149]. Due to a possible causal role of aberrant gut microbiota in lupus pathogenesis, FMT of feces from healthy individuals might be beneficial in patients with lupus and a clinical trial of FMT in patients with active SLE effectively modified the gut microbiota [150]. More studies on this topic are ongoing [151,152,153,154]. On the other hand, the strategies of the direct attenuation of intestinal integrity by the protective molecules from probiotics (such as the epithelial cell growth factors and short-chain fatty acids) with neutralization of PAMPs from gut translocation are also interesting. Surprisingly, an alteration of free fatty acids (FFA) in serum and feces of patients with lupus is common and partially associated with gut dysbiosis [72]. The administration of butyrate in MRL/lpr lupus-prone mice ameliorates gut microbiota dysbiosis [155] which might be even more interesting than probiotics due to the possible easier preparation and storage of FFA as a chemical drug. Nevertheless, probiotics are the current strategy to improve gut integrity and gut dysbiosis with the most intensive studies with an adverse effect (systemic dissemination) in patients with potent immunosuppression [156]. However, the disseminated probiotics are easily eradicated by a simple antibiotic because of the facultative anaerobic nature of the organisms. Hence, the additional indicators of gut dysbiosis, gut leakage, and the monitoring biomarkers might be beneficial for lupus. More studies are warranted.

## 9. Conclusions

The summary of our working hypothesis is presented in Figure 3. As such, gut leakage caused by immune-complex deposition, gut dysbiosis, infection, and medications might be a silent exacerbating factor of lupus activity. Gut translocation of PAMPs, especially LPS and BG, into blood circulation may induce systemic inflammation and enhance lupus activity through induction of several cytokines and several types of cell death from innate immune cells. Moreover, the molecular mimicry between gut pathogens and self-antigens might possibly increase the autoantibodies and lupus flare-ups. Because probiotics could attenuate gut dysbiosis and strengthen gut integrity without serious adverse effects, probiotic treatment in the selected patients with leaky gut in lupus using several biomarkers is interesting.

## Figures and Tables

**Figure 1 ijms-23-08223-f001:**
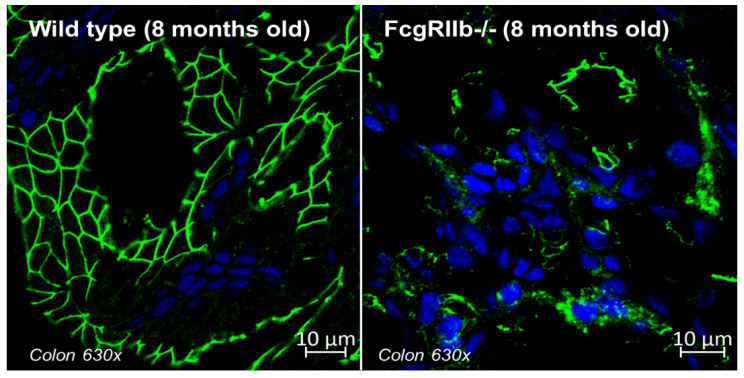
The representative picture of zona occludens-1 (ZO-1, green fluorescent color), a tight junction protein, in enterocyte of the colon from 8-week-old mice of wild-type control and FcgRIIb-/- lupus mice are demonstrated (original magnification 630×) [26]. The blue color is the staining of the nucleus using DAPI (4′,6-diamidino-2-phenylindole), a blue-fluorescent DNA stain (Alexa Fluor 488, Abcam, Cambridge, MA, USA). The samples were prepared in Cryogel (Leica Biosystems, Richmond, IL, USA) and photographed by a ZEISS LSM 800 (Carl Zeiss, Germany). Notably, there is a well-defined green color border in wild type versus the unclear borders of ZO-1-stained colon in FcgRIIb-/- lupus mice at 8 weeks old (full-blown lupus).

**Figure 2 ijms-23-08223-f002:**
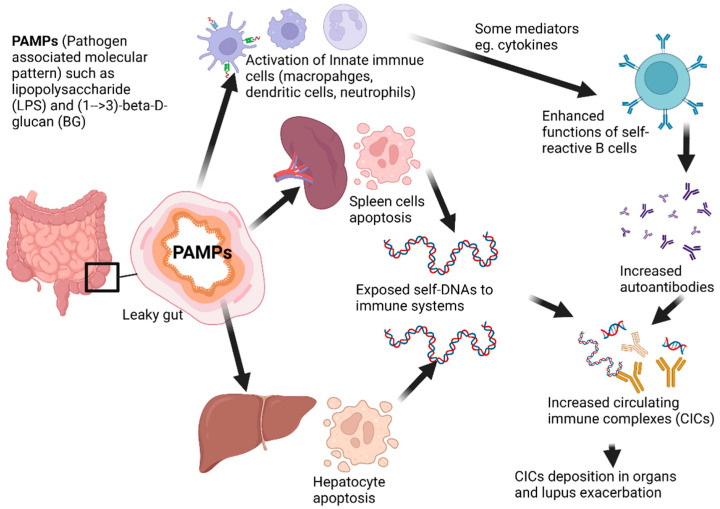
The working hypothesis diagram demonstrates the possible mechanism of leaky gut-exacerbated lupus through the elevated circulating immune complexes, from (i) increased exposure to the self-antigens from overwhelming apoptosis (such as splenocytes and hepatocytes) and (ii) enhanced autoantibody production from inflammatory mediators, due to the activation by PAMPs from leaky gut.

**Figure 3 ijms-23-08223-f003:**
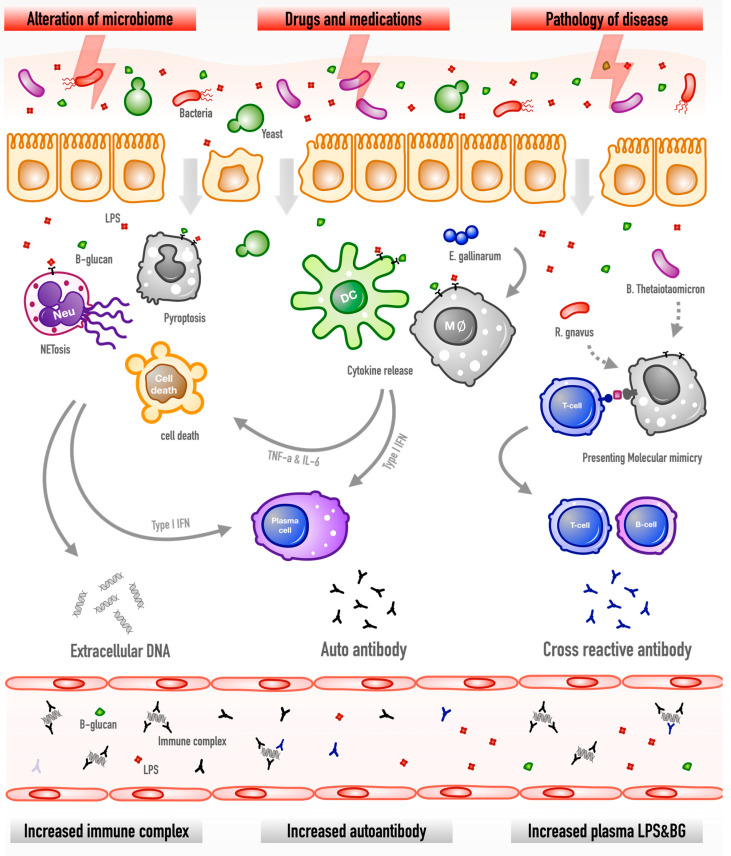
The hypothesis diagram demonstrates gut permeability defects in lupus through several mechanisms (microbiome alteration, drugs, and lupus-induced inflammation) facilitate the translocation of microbial molecules that induce the death of immune cells (apoptosis, NETosis, and pyroptosis), the release of cytokines (type I IFN and IL-6), and molecular mimicry, results in extracellular DNA exposure, autoantibody production, and cross-reactive antibodies. Hence, leaky gut might enhance pathogen molecules in the blood, such as lipopolysaccharides (LPS) from Gram-negative bacteria and (1→3)-β-d-glucan (BG) from gut fungi, autoantibody production, and the circulating immune complex.

**Table 1 ijms-23-08223-t001:** Beneficial probiotics in SLE.

Probiotic	Effect To Attenuate Leaky Gut	Studying in SLE Human or Lupus Murine Model		Ref.
		Strain	Models/Observed Effect	
*Lactobacillus rhamnosus*	- Secrete metabolites to contribute intestinal homeostasis and barrier function [157,158] - Improve gut permeability and modulate gut dysbiosis [159,160] - Prevent DNBS- [160] and DSS- [161] induced colitis	GG ATCC 9595	Pristine-induced murine model: anti-dsDNA ↓, IFN*γ* ↓, Th1-Th17 polarization ↓, T_reg_ cell ↑	[162,163]
			Patients with SLE: miR-181a ↓, miR155 ↓	[154]
		LMS 201	MRL/lpr murine model: attenuate lupus nephritis, IL-6 ↓, IL-10 ↑, IgG2a ↓	[58]
*Lactobacillus delbrueckii*	- Promote intestinal integrity and upregulate tight junction protein in LPS-challenged piglets [164]	PTCC 1743	Pristine-induced murine model: anti-dsDNA ↓, IFN*γ* ↓, Th1-Th17 polarization ↓, T_reg_ cell ↑	[154,163]
*Lactobacillus Plantarum*	- Prevent spontaneous colitis in IL-10 knockout mice [165] - Prevent methotrexate induce enterocolitis [166] - Attenuates phorbol ester-induced redistribution of ZO-1 and occludin in vivo [167] - Improve gut dysbiosis and oxidative status in diabetic rats [168]	LP299v	NZB/W F1 murine model: anti-inflammatory phenotype of BM-DCs ↑, IL-10 ↑, IL-12 ↑	[169]
		LC40	NZB/W F1 murine model: improve gut dysbiosis, renal injury ↓, plasma LPS ↓, occludin and ZO-1 expression ↑, TNF-*α* ↓, Th1-Th17 polarization ↓, T_reg_ cell ↓	[74]
*Lactobacillus reuteri*	- Prevent methotrexate induce enterocolitis [166] - Improve gut dysbiosis and prevent gut barrier disruption in antibiotic-induced leaky gut [170] - Increase production of indole and IL-22 in lumen [171]	GMNL-263	NZB/W F1 murine model: IL-6 ↓, TNF-*α* ↓, MMP-9 ↓, CRP ↓, TLR-4 ↓, TLR-7 ↓, TLR-9 ↓, Treg cell ↑	[153,172]
		GMNL-89	NZB/W F1 murine model: hepatic apoptosis ↓, IL-6 ↓, TNF-*α* ↓, MMP-9 ↓, CRP ↓	[153]
		DSM 17509	NZB/W F1 murine model: survival rate ↑, anti-inflammatory phenotype of BM-DCs ↑, IL-10 ↑, IL-12 ↑	[169]
*Lactobacillus Casei*	- Prevent TNF-*α* induced epithelium dysfunction in vitro [173] - Reduce severity of DSS-induced colitis [174]	B255	NZB/W F1 murine model: survival rate ↑, anti-inflammatory phenotype of BM-DCs ↑, IL-10 ↑, IL-12 ↑	[169]
		shirota	MRL/lpr murine model: B220 positive T cell in spleen and MLN ↓, IL-6 expression in macrophage ↓	[175]
*Lactobacillus* *Paracasei*	- Prevent age-related leaky gut and inflammation [176]	GMNL-32	NZB/W F1 murine model: cardiac apoptosis ↓, hepatic apoptosis ↓, IL-6 ↓, TNF-*α* ↓	[153,177]

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
