# Peer review of "Gut Barrier Damage and Gut Translocation of Pathogen Molecules in Lupus, an Impact of Innate Immunity (Macrophages and Neutrophils) in Autoimmune Disease"

_ijms, 2022, doi:10.3390/ijms23158223_

Round 1
Reviewer 1 Report
The manuscript intends to summarize a relative new developing field in lupus pathogenesis and disease progression. 1. The manuscript is not well organized, therefore, delivering a mixed/confusing message. It covers all parts but lacking focus. ie. immune complex deposition in GI track cause gut leakage, which in turn leads to lupus flare? This is unlikely, because immune complex deposition is continuously happening; 2. Gut leakage induce cell death that leads to lupus exacerbation. This is also unlikely, as it's well known that lupus patients exhibit defects of apoptotic cell clearance and excessive apoptotic bodies found in lupus patients long before lupus clinical manifestation. Those are continue contributing factors to lupus. 3. The cytokine section is very much under developed, only two cytokines were mentioned. Many other parts talked about cytokines as well. 4. Are there known genetic associated gut leakage (but dysbiosis) in lupus? as lupus is a disease caused by genetic and environmental factors. 5. Suggest to put a separate section for therapeutics targeting gut leakage, ie. probiotics, feces administration, food/nutrition, et al.
Minor issues:
1. Several abbreviations were not explained at the first place.
2. Some terms are not consistent, TNFa vs TNF-a.
3. Factors (immune complex, drug, et al) lead to gut leaking should be included in Figure 2
Author Response
The manuscript intends to summarize a relative new developing field in lupus pathogenesis and disease progression.
- The manuscript is not well organized, therefore, delivering a mixed/confusing message. It covers all parts but lacking focus. ie. immune complex deposition in GI track cause gut leakage, which in turn leads to lupus flare? This is unlikely, because immune complex deposition is continuously happening;
ANS: We apologize for the unclear presentation. We tone down this hypothesis and explain more. Although CIC is continuously happening, there might be an increase and decrease CICs as we can see lupus flare-up and lupus inactivity. Theoretically, the increased CICs might be due to an increase in self-antigens and/ or enhanced auto-antibody production. In our hypothesis, we proposed that circulating pathogen molecules from gut leakage induces injury of immune cells, especially spleen apoptosis, and the failure of apoptotic body clearance from an overwhelming apoptosis increases self-antigens and finally elevates CICs. Then, we wrote a short version of this hypothesis with a new figure in the new version manuscript.
- Gut leakage induce cell death that leads to lupus exacerbation. This is also unlikely, as it's well known that lupus patients exhibit defects of apoptotic cell clearance and excessive apoptotic bodies found in lupus patients long before lupus clinical manifestation. Those are continue contributing factors to lupus.
ANS: We thank the reviewer for the comment. We tune down this hypothesis and explain more. Because the defects of apoptotic cell clearance and excessive apoptotic bodies are one of the well-known lupus pathogenesis, we hypothesize that further apoptosis might increase CIC and exacerbate lupus activity. Because chronic inflammation could induce spleen apoptosis in several models, including DSS-induced leaky gut, leaky gut might be one of the conditions that enhance apoptosis and exacerbate lupus. Then, we wrote a short version of this hypothesis with a new figure in the new version manuscript.
- The cytokine section is very much under developed, only two cytokines were mentioned. Many other parts talked about cytokines as well.
ANS: We thank the reviewer for the comment. We change the name of the section to “Leaky gut-induced inflammation and molecular mimicry in lupus” because cytokines are not the real main theme of the section and “inflammation” might be more proper.
- Are there known genetic associated gut leakage (but dysbiosis) in lupus? as lupus is a disease caused by genetic and environmental factors.
ANS: We thank the reviewer for the comment. The host genes might allow some microbiota to survive which is possible in hereditary lupus from complement defects. However, that environmental factors are more important than host genetics in shaping the human gut microbiota. Then, we explain more on this in the topic “gut dysbiosis in lupus” of the new version manuscript.
- Suggest to put a separate section for therapeutics targeting gut leakage, ie. probiotics, feces administration, food/nutrition, et al.
ANS: We thank the reviewer for this suggestion and include this part accordingly.
Minor issues:
- Several abbreviations were not explained at the first place.
ANS: We apologize for the unclear presentation and correct them accordingly.
- Some terms are not consistent, TNFa vs TNF-a.
ANS: We apologize for the unclear presentation and correct them accordingly.
- Factors (immune complex, drug, et al) lead to gut leaking should be included in Figure 2
ANS: We apologize for the unclear presentation and correct them accordingly.
Reviewer 2 Report
In this review, the authors discussed gut barrier damage and gut translocation of pathogen molecules in lupus. Totally, the work is a significant contribution to the field.
Its will be better to shorten the non-related content and foucus on the deatils of lupus.
Author Response
In this review, the authors discussed gut barrier damage and gut translocation of pathogen molecules in lupus. Totally, the work is a significant contribution to the field.
Its will be better to shorten the non-related content and focus on the details of lupus.
ANS: We thank the reviewer for the comment and explain more to link the topic into lupus.